# I-Max: Maximize the Resolution Potential of Pre-trained Rectified Flow Transformers with Projected Flow

## Abstract

Rectified Flow Transformers (RFTs) offer superior training and inference efficiency, making them likely the most viable direction for scaling up diffusion models. However, progress in generation resolution has been relatively slow due to data quality and training costs. Tuning-free resolution extrapolation presents an alternative, but current methods often reduce generative stability, limiting practical application. In this paper, we review existing resolution extrapolation methods and introduce the I-Max framework to maximize the resolution potential of Text-to-Image RFTs. I-Max features: (i) a novel Projected Flow strategy for stable extrapolation and (ii) an advanced inference toolkit for generalizing model knowledge to higher resolutions. Experiments with Lumina-Next-2K and Flux.1-dev demonstrate I-Max's ability to enhance stability in resolution extrapolation and show that it can bring image detail emergence and artifact correction, confirming the practical value of tuning-free resolution extrapolation.

## 1 Introduction

Over the past few years, diffusion model Sohl-Dickstein et al. (2015) has made significant breakthroughs across various dimensions, including diffusion process Ho et al. (2020); Song et al. (2020a;b); Liu et al. (2022); Lipman et al. (2022), model design Ho et al. (2022); Rombach et al. (2022); Teng et al. (2023), network architecture Peebles & Xie (2023), *etc*, and yielding numerous practical applications. Building on the accumulated experience from these explorations, rectified flow transformers (RFTs) Ma et al. (2024a) have now been recognized as a potential future direction for scaling up diffusion models, leading to the development of successful open-source text-to-image models like Stable Diffusion 3 Esser et al. (2024), Lumina-T2X Gao et al. (2024), and Flux Black Forest Labs (2024). Although these models have achieved significant improvements in various aspects like generation quality, aesthetics, and text-image alignment, their native generative resolution has remained limited to the $1024^2$-$2048^2$ range. This restricts the direct application of AI-generated visual content in scenarios with high-quality demand.

However, directly training high-resolution generative models is currently less practical, considering the high data quality requirements and the exponential increase in training costs associated with ultra-high-resolution generation. In fact, training a low-resolution diffusion model already requires dozens of days on hundreds of GPUs Podell et al. (2023). Therefore, some existing works achieve a balance by supervised fine-tuning – improving the model's generative resolution via tuning on a certain amount of high-resolution samples Cheng et al. (2024); Chen et al. (2024); Ren et al. (2024). While another line of research focuses on directly hacking the model's inference stage to achieve tuning-free resolution extrapolation – this allows for high-resolution generation almost as a "free lunch", but it also significantly decreases the model's generative stability. In this paper, based on pre-trained text-to-image RFTs, we discover that rectified flow not only facilitates efficient inference but also offers inherent advantages in resolution extrapolation. Combined with the empirical knowledge of transformer's context length extrapolation, we propose the I-Max framework, which enables stable generation across a range of extrapolated resolutions, enhancing the practical application value of tuning-free resolution extrapolation.

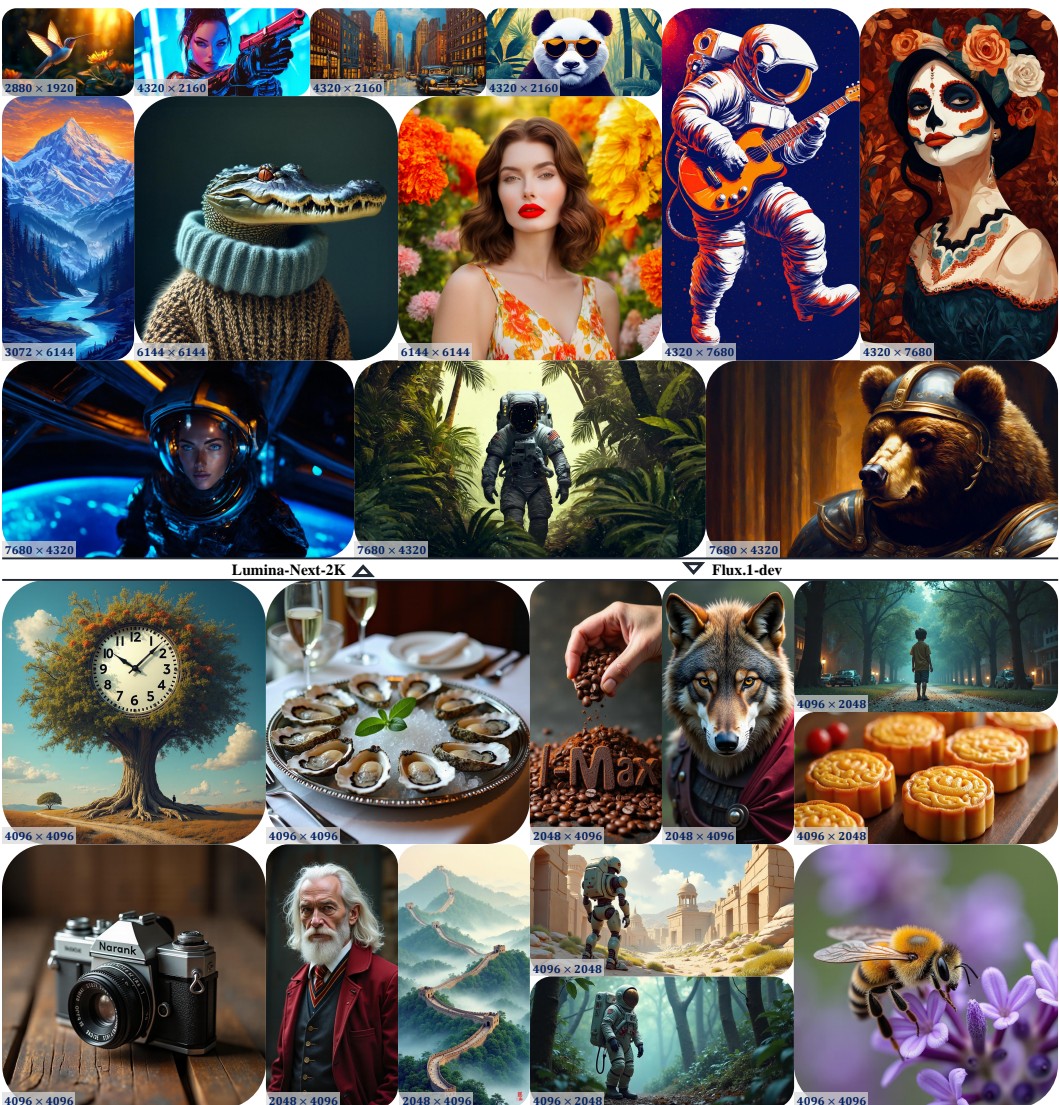

Figure 1: Landscape images crafted by Lumina-Next-2K and Flux.1-dev equipped with I-Max.

Reviewing the existing works on resolution extrapolation, we first decouple the challenges they address into two perspectives: (i) *"how to guide?"*, which refers to how to use reliable native-resolution generation results to guide high-resolution generation, and (ii) *"how to infer?"*, which is about enhancing the model's generative ability to generalize better to extrapolated resolutions during the inference phase. We consider both perspectives equally essential and build the I-Max framework upon them. However, improving the model's generalization to extrapolated resolutions seems quite intuitive, while using low-resolution generation for guidance may not appear strictly necessary. For this, we demonstrate through experiments that the diffusion model's generalization ability to the extrapolated generation resolutions varies over time. It exhibits weaker extrapolation capability during the early denoising stages when generating coarse image content from random noise, but demonstrates a more robust ability during the later stages of denoising the coarse content (please refer to Sec. 3.5). In the following paragraphs, we will summarise how existing works address the questions of *"how to guide?"* and *"how to infer?"*, and introduce our design choices in I-Max that are tailored for RFTs.

To solve *"how to guide?"*, a direct approach is to upsample the low-resolution results, add a certain amount of noise, and then denoise back, similar to SDEdit Meng et al. (2021). DemoFusion Du et al. (2024) improves this two-stage paradigm by introducing the concept of "skip-residual", which

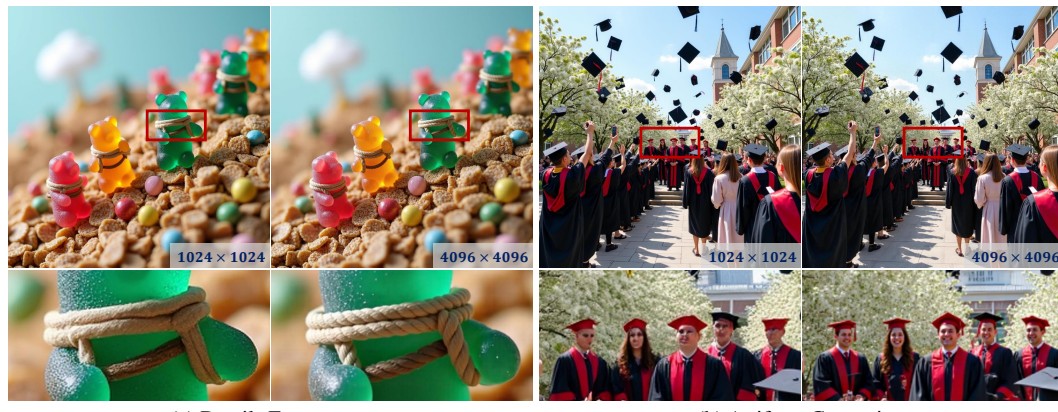

(a) Details Emergence                    (b) Artifacts Correction

Figure 2: Extrapolation of 1K→4K on Flux.1-dev enhances the generated result with richer local details and corrects artifacts in small objects.

integrates the noise-injected low-resolution embeddings at each timestep in a time-conditioned manner. Although this "skip-residual" method is simple and effective, it introduces fixed noise at each timestep, which limits its ability to adjust the subtle change of denoising direction. On the contrary, Upsample Guidance Hwang et al. (2024) downsamples the high-resolution embeddings to the native resolution to infer the simultaneous low-resolution guidance. Although it carefully balances the distribution shift of embeddings and noises caused by downsampling, it still fails to maintain the model performance at the native resolution. Another more direct approach is integrating low-resolution generation and high-resolution extrapolation into a single denoising process in a relay manner Teng et al. (2023); Zhuo et al. (2024), but it only improves efficiency. In I-Max, we try to explore the optimal solution under the context of the rectified flow model. Particularly, we treat the low-resolution space as a low-dimensional projection of the high-resolution space, which means the low-resolution flow can be regarded as the projection of the ideal high-resolution flow. And considering the linear interpolation characteristic of rectified flow – where the optimal direction of the current flow can be directly constructed as the vector from the current position to the endpoint, we can easily build dynamic guidance in the projected space at each timestep, which we term Projected Flow (see details in Sec. 2.2).

As for *"how to infer?"*, whether UNet or Transformer-based diffusion models have been found to lack good generalization to extrapolated resolutions – direct extrapolation often leads to the collapse of the generated content. A direct solution to this can be found in MultiDiffusionBar-Tal et al. (2023) and its subsequent work Du et al. (2024); Haji-Ali et al. (2024); Lin et al. (2024a;b), which sample overlapping patches at the training resolution to ensure stable inference. In contrast, some other approaches He et al. (2023); Zhang et al. (2023); Huang et al. (2024) adopt dilated convolution to extend the model's perception field while doing resolution extrapolation. However, these methods are only practical for CNN-based diffusion models and cannot be applied to the DiT architecture. For this, FiT Lu et al. (2024) and Lumina-T2X Gao et al. (2024) draw on the successful experience of long-text extrapolation in LLMs and adopt NTK-aware Scaled RoPE loc (2024) to achieve direct extrapolation in a limited range (*e.g.*, $256^2 \rightarrow 512^2$ or $1024^2 \rightarrow 2048^2$). In addition to adjusting models' spatial perception, the SNR shift Esser et al. (2024) and attention entropy shift Jin et al. (2024) are also found to be critical to resolution extrapolation and need to be corrected. This paper identifies the essential inference techniques for RFTs and integrates them as an inference toolkit into the I-Max framework (see details in Sec. 2.3). This ensures that the model can generate expressive, detailed content at extrapolated resolutions.

To validate the effectiveness of I-Max, we tune Lumina-Next Zhuo et al. (2024) with self-collected high-quality, high-resolution images and obtain Lumina-Next-2K, a native 2K generation RFT. On Lumina-Next-2K, I-Max can achieve 4× to 16× resolution extrapolation, directly generating images up to 8K resolution, as shown in Fig. 1. We also validate I-Max on Flux.1-dev Black Forest Labs (2024), where it achieve stable 4K image generation, demonstrating its general applicability to MMDiT structured RFT models. Notably, as shown in Fig. 2, even though Flux.1-dev can generate ultra-high-quality images, conducting resolution extrapolation through I-Max still provides benefits,

such as local detail emergence and correcting artifacts like small faces in crowdy scenes. Moreover, the overall improvement in generation quality suggests the existence of untapped potential in the model's inference phase. In the field of large language model (LLM) research, there is growing awareness of the potential benefits of increasing inference costs, leading to the concept of the "inference scaling law" Snell et al. (2024). We hope that our work can (to some extent) help similar advancements occur in large vision models.

## 2 METHODOLOGY

Here, we will introduce the core components of I-Max in the following order: in Sec. 2.1, we first introduce the preliminary knowledge about rectified flow; in Sec. 2.2, we introduce how we achieve low-resolution guidance via projected flow which is tailored for rectified flow models, and in Sec. 2.3, we further summary the inference techniques that are essential for DiTs to infer at extrapolated resolutions.

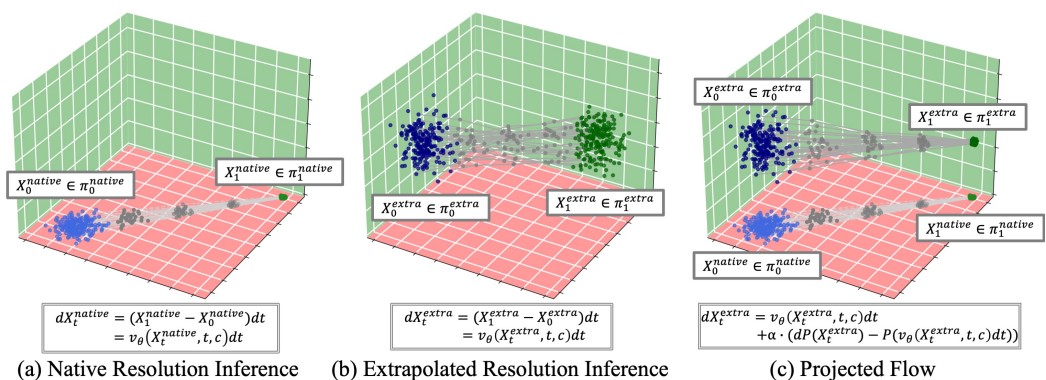

(a) Native Resolution Inference    (b) Extrapolated Resolution Inference    (c) Projected Flow

Figure 3: Illustration of Projected Flow Mechanism:(a) In the low-dimensional space of native resolution, RFTs accurately predict the flow direction, ensuring precise distribution transfer. (b) In the high-dimensional space of extrapolated resolution, RFTs struggle to accurately predict the flow direction, degrading the quality of distribution transfer. (c) Projected Flow treats the flow in the low-dimensional space as a deterministic projection of the flow in the high-dimensional space, reducing the difficulty of predicting the flow direction at extrapolated resolutions.

### 2.1 PRELIMINARY: RECTIFIED FLOW

Rectified Flow Liu et al. (2022) and Flow Matching Lipman et al. (2022) attempt to simplify the construction of ODE models by linearly interpolating between two distributions. Here, we introduce the preliminary knowledge in the context of Rectified Flow. Given samples $X_0 \in \pi_0$ and $X_1 \in \pi_1$, where $\pi_0$ is the noise distribution and $\pi_1$ is the image distribution in this paper. Rectified flow builds the path from $X_0$ to $X_1$ as a linear flow with the direction of $(X_1 - X_0)$, and the intermediate state $X_t$ can be represented by $X_t = tX_1 + (1 - t)X_0$. Therefore, we can build the ODE of $X_t$ as

$$dX_t = (X_1 - X_0)dt. \tag{1}$$

Considering that during denoising, the ODE is *non-causal* – since $X_1$ is the unknown target – we can use $v_\theta(X_t, t, c)$ to fit in the direction $(X_1 - X_0)$ constructing the neural ODE model, where $c$ is additional conditions like text and class labels. In this way, we can transfer $X_0$ to $X_1$ by the prediction $d\hat{X}_t = v_\theta(X_t, t, c)dt$.

### 2.2 PROJECTED FLOW

Neural ODE models built upon rectified flow (*e.g.*, RFTs) exhibit excellent training and inference efficiency, and they can provide reliable direction predictions $d\hat{X}_t^{\text{native}} = v_\theta(X_t^{\text{native}}, t, c)$ at the model's native training resolution as illustrated in Fig. 3 (a). And when it comes to resolution extrapolation, here we first put forward the core understanding in this paper that – *the low-resolution image space is a low-dimensional subspace of the high-resolution image space, and every image /*

*flow in the high-resolution space has a corresponding projection in the low-resolution space*. From this perspective, an ideal resolution extrapolation ability can be expressed as the model's "projection invariance", *i.e.*,

$$P(v_\theta(X_t^{\text{extra}}, t, c)) = v_\theta(P(X_t^{\text{extra}}), t, c), \tag{2}$$

where $X_t^{\text{extra}}$ is the intermediate state at the extrapolated resolution and $P(\cdot)$ is a projection function. However, as we discussed earlier, without any optimization, RFTs do not demonstrate strong generalization capabilities at extrapolated resolutions. In Fig. 3 (b), we illustrate this as an inaccurate distribution transfer resulting from incorrect direction prediction in the extrapolated-resolution space.

To address this issue, we propose the idea of Projected Flow with an intuitive working mechanism – we first predict the flow at the native resolution and then guide the high-resolution space's flow to follow the deterministic projection in the lower-dimensional space, as illustrated in Fig. 3 (c). This approach allows the model to focus only on the additional extrapolated information in the high-resolution space, significantly enhancing the stability of the resolution extrapolation. In other words, with the assumption that the low-resolution prediction $\hat{X}_1^{\text{native}}$ is an approximation of $X_1^{\text{extra}}$ in the low-dimensional space, *i.e.*, $P(X_1^{\text{extra}}) \approx \hat{X}_1^{\text{native}}$, the ODE at extrapolated resolution $dX_t^{\text{extra}} = (X_1^{\text{extra}} - X_0^{\text{extra}})dt$ becomes *partially-causal* as we know the low-dimensional projection of $X_1^{\text{extra}}$. Following the definition of Rectified Flow, the ideal ODE of the low-dimensional component can be simulated by:

$$dP(X_t^{\text{extra}}) = (P(X_1^{\text{extra}}) - P(X_0^{\text{extra}}))dt \tag{3}$$

$$\approx (\hat{X}_1^{\text{native}} - P(X_0^{\text{extra}}))dt. \tag{4}$$

Furthermore, the linear characteristic of rectified flow allows us to adjust the direction at any given time $t$ according to the deterministic endpoint:

$$dP(X_t^{\text{extra}}) \approx \frac{(\hat{X}_1^{\text{native}} - P(X_t^{\text{extra}}))}{1 - t} dt. \tag{5}$$

Here, we balance the magnitude of $dP(X_t^{\text{extra}})$ at different timestep $t$ using a scaling factor $1 - t$. Afterwards, to encourage the high-resolution flow to follow the low-dimensional projection, we build low-resolution guidance via projected flow on $v_\theta(X_t^{\text{extra}}, t, c)$ in the form of classifier-free guidance Ho & Salimans (2022) as

$$\tilde{v}_\theta(X_t^{\text{extra}}, t, c) = v_\theta(X_t^{\text{extra}}, t, c) + \alpha_t \cdot \left( \frac{dP(X_t^{\text{extra}})}{dt} - P(v_\theta(X_t^{\text{extra}}, t, c)) \right) \tag{6}$$

$$= v_\theta(X_t^{\text{extra}}, t, c) + \alpha_t \cdot \left( \frac{(\hat{X}_1^{\text{native}} - P(X_t^{\text{extra}}))}{1 - t} - P(v_\theta(X_t^{\text{extra}}, t, c)) \right). \tag{7}$$

Notably, using a fixed $\alpha_t$ throughout the denoising process may limit the additional details brought by extrapolation. Thus, we implement a cosine decay strategy following Du et al. (2024) as $\alpha_t = 1 + 0.5 \cdot cos(\pi t)$. As for the projection function, we utilize a low-pass filter to simulate the projection onto the low-dimension space while maintaining the size of the features.

## 2.3 INFERENCE TOOLKIT

Beyond the projected guidance, we introduce additional inference techniques tailored for RFT that can enhance the model's ability to generalize to extrapolated resolutions. Note that some parameter settings need to reference the base model's *native resolution*. However, Flux.1-dev is a multi-resolution generative model trained within the $256^2$ to $2048^2$ range, so it does not have a clearly defined *native resolution*. In this case, we refer to $1024^2$ as the native resolution for calculations and introduce an additional re-scaling hyper-parameter.

**Denoise beyond the training resolution.** 2D rotary position embedding (RoPE) Su et al. (2024) is widely adopted by DiTs to model 2D positions. It encodes position information of each axis using a frequency matrix $\Theta = \text{Diag}(\theta_1, \cdots, \theta_d, \cdots, \theta_{d_{\text{head}}/4})$ with $\theta_d = b^{-4d/d_{\text{head}}}$, where $b$ is the rotary base. Despite its strong generalization across various sequence lengths, it still struggles to generalize beyond the sequence lengths encountered during the training phase. To address this, NTK-aware scaled RoPE loc (2024) is proposed to solve the zero-shot context length extrapolation

problem in LLMs by rescaling the rotary base $b$ as $b' = b \cdot s$. Here, $s = \sqrt{\frac{L^{\text{extra}}}{L^{\text{native}}}}$ is the scaling factor, where $L^{\text{extra}}$ and $L^{\text{native}}$ is the sequence length at the native resolution and the extrapolated resolution, respectively. Moreover, it has also been proven effective for DiTs Lu et al. (2024); Gao et al. (2024). Therefore, in this paper, we also applied NTK-aware scaled RoPE in I-Max to ensure that the model maintains accurate spatial modelling capabilities during extrapolation. In particular, we take $b' = b \cdot s$ for Lumina-Next-2K and $b' = 2.5 \cdot b \cdot s$ for Flux.1-dev.

**Balance the SNR shift.** Previous arts Hwang et al. (2024); Esser et al. (2024) point that, given a defined diffusion process, the signal-noise-ratio (SNR) of $X_t$ is resolution-dependent. And in the context of rectified flow model, for a constant image $X_1$, the SNR of images with different resolutions can be formulated as $\sigma(X_t^{\text{extra}}, t) = s^2 \cdot \sigma(X_t^{\text{native}}, t)$. Such a shift in image SNR will also degrade model performance on extrapolated resolutions. Therefore, Gao et al. (2024) re-shift the time $t$ of RFTs when denoising to balance the SNR shift, $t^{\text{extra}} = \frac{t^{\text{native}}}{s^* - s^* \cdot t^{\text{native}} + t^{\text{native}}}$, where $s^*$ is a hyper-parameter as we cannot calculate the exact SNR shift for unknown images. In this paper, we keep the time-shifting operation and set $s^* = s$ for Lumina-Next-2K and $s^* = 1.5 \cdot s$ for Flux.1-dev.

**Balance the entropy shift of self-attention.** When performing resolution extrapolation, the sequence length of the transformer processes increases exponentially with the resolution. Longer sequences significantly increase the entropy of the self-attention scores, affecting the information aggregation process. To adaptively balance the attention distribution, Jin et al. (2024) rewrite self-attention as $Attention(Q, K, V) = softmax(s \cdot \frac{QK^T}{\sqrt{d}} \cdot V)$. We adopt proportional attention for both Lumina-Next-2K and Flux.1-dev.

**Balance the image/text sequence length ratio for MMDiT.** In methods based on the MMDiT architecture, self-attention is performed on the joint sequence of text and image tokens. The number of text tokens is usually fixed, but during resolution extrapolation, the number of image tokens increases exponentially, significantly changing the ratio of image to text tokens within the joint sequence. This results in a noticeable shift in the proportion of image and text information each token receives during the self-attention process. In I-Max, we find that simply repeating the text tokens to match the image sequence's extrapolation improves the quality of generated images, *e.g.*, for a scaling factor $s$, we repeat the text sequence for $s^2$ times as $c = [c, c, \cdots, c]$. Additionally, considering that Flux.1-dev applies the same $(0, 0, 0)$ position index to all text tokens, to prevent out-of-distribution values in the relative positions between image and text tokens, we add grid position indexes to the repeated text tokens. In the following sections, we refer to this operation as text duplication. We only adopt this strategy for Flux.1-dev since Lumina-Next-2K utilizes cross-attention architecture for text injection.

## 3 EXPERIMENTS

### 3.1 IMPLEMENTATION DETAILS

In this paper, we use a self-trained Lumina-Next-2K model and the open-source Flux.1-dev Black Forest Labs (2024) model as representative rectified flow transformers (RFTs) to validate the general effectiveness of I-Max for RFTs. Lumina-Next-2K is a 2K generative model derived from the open-source Lumina-Next model Zhuo et al. (2024), after 60,000 iterations of supervised fine-tuning on 800K self-collected high-quality data of the $2048^2$ resolution. Flux.1-dev, on the other hand, is a multi-resolution generative model capable of producing images with resolutions ranging from $256^2$ to $2048^2$ pixels.

In terms of image evaluation, as mentioned in previous work, existing metrics are not well-suited for high-resolution image evaluation because they require downsampling the images. Additionally, considering the efficiency of generating high-resolution images, it is challenging to produce enough test images (typically in tens of thousands) for reliable metric calculation in each experiment. Therefore, human evaluation remains the most reliable method. In Pixart-$\Sigma$ Chen et al. (2024), AI preference has already been shown to have a high degree of consistency with human preference. Considering the cost and efficiency of human evaluation, in this paper, we use multi-model large language mod-

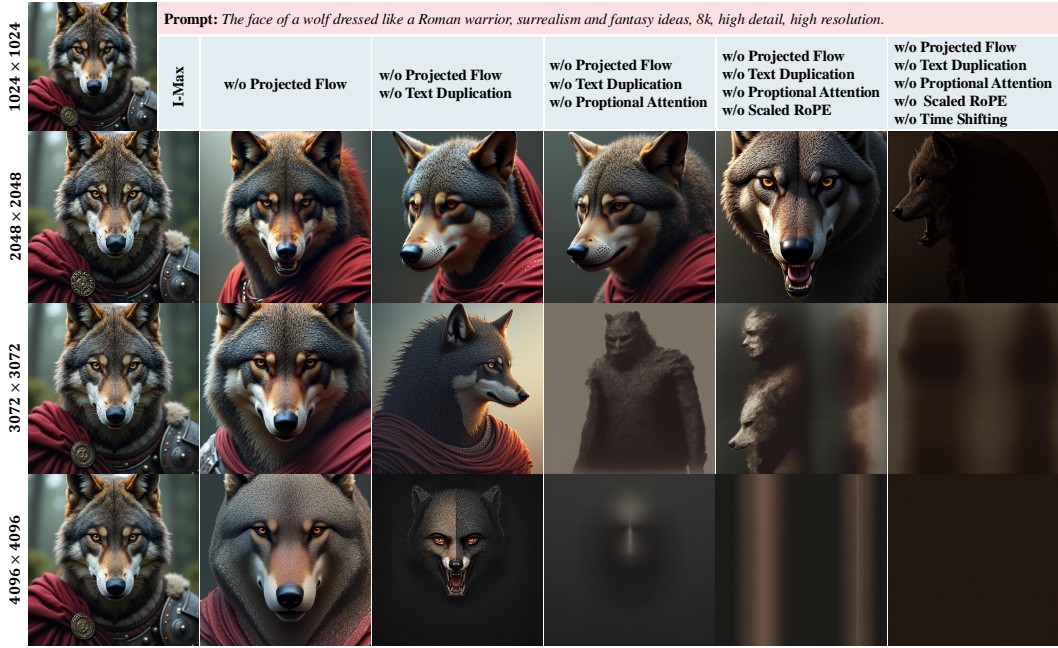

Figure 4: Sequential ablation of I-Max. We illustrate the effect of sequentially removing different components of I-Max on Flux.1-dev across the 1K→4K resolution range.

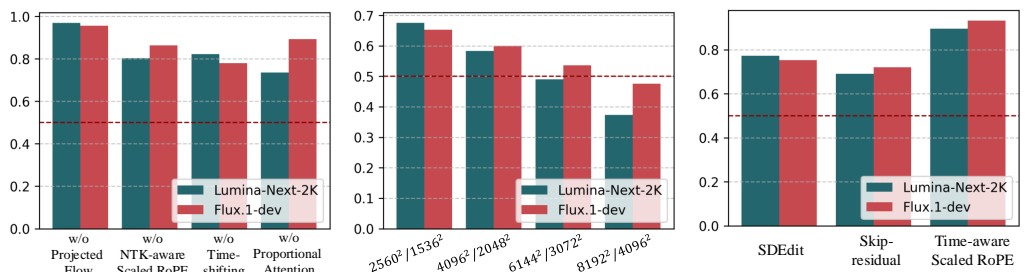

Figure 5: Single ablation results of I-Max.

Figure 6: Performance gains at different resolutions.

Figure 7: Comparison of low-resolution guidance techniques.

els (e.g., GPT-4o OpenAI (2023)) to compare image quality according to the prompts, serving as a low-cost alternative to human evaluation. The example of use case can be found in Appendix A.1.

## 3.2 EFFECTIVENESS OF EACH COMPONENT OF I-MAX

In Fig. 4, we present the results of sequential ablation across the 1K→4K resolution range on Flux.1-dev. We observe that Projected Flow significantly ensures inference stability at extrapolated resolutions, preserving the global structural integrity within the image. After removing Projected Flow, the subsequent results indicate the model's ability to generalize at the extrapolated resolution, which directly affects whether the model can produce meaningful local details during resolution extrapolation. These results also confirm that as the resolution increases, the model's generalization ability degrades significantly. However, the inference toolkit we introduced effectively prevents model collapse. Even for resolutions like $2048^2$, which the Flux.1-dev model claims to handle, leveraging additional inference techniques improves the quality of the generated results. A sequential ablation study based on Lumina-Next-2K can be found in Appendix A.2. Although the model collapse mode may vary due to differences in the DiT architecture, the effectiveness of each module is consistently validated.

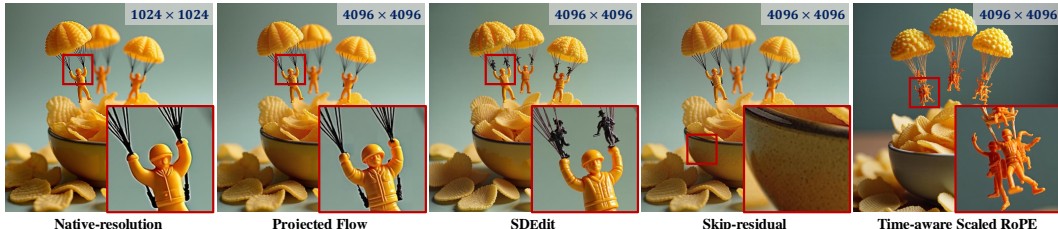

Figure 8: Illustration of results obtained by different low-resolution guidance approaches.

Additionally, in Fig. 5, we verify the necessity of each module through single ablation. We randomly generate 300 images at the $4096 \times 4096$ resolution using Lumina-Next-2K and Flux.1-dev for each setting and conduct a comparison using GPT-4o preference voting between w/ and w/o a specific module. The experimental results show that I-Max achieve over $50\%$ win rates under all settings, demonstrating that the absence of any individual component leads to a degradation in overall performance.

### 3.3 GAINS FROM RESOLUTION EXTRAPOLATION

Existing resolution extrapolation methods, while capable of producing some impressive high-resolution images, often reduce the stability of the generative model. This, in turn, increases hidden costs in practical applications, making these methods less practical. In this section, using GPT-4o preference assessments, we compare the model's generation results at its native resolution with high-resolution images generated using I-Max. As shown in Fig. 6, we can observe that for a certain range of scaling factors, the extrapolated results even achieve over $50\%$ win rates – indicating that, in most cases, I-Max can provide positive gains to the overall quality of the generated results. We attribute this to the stability provided by Projected Flow, which ensures that I-Max can maximize the model's resolution potential. However, as the scaling factor continues to increase, we see the win rate gradually drop below $50\%$. This is mainly because the model's prior knowledge is insufficient to supplement meaningful content for an indefinitely increasing number of pixels, indicating a potential upper limit.

### 3.4 COMPARISON OF DIFFERENT LOW-RESOLUTION GUIDANCE

In I-Max, we propose Projected Flow, specifically tailored to the characteristics of rectified flow, which leverages low-resolution generation results to improve the stability of high-resolution generation. Here, we compare Projected Flow with existing low-resolution guidance methods to demonstrate its superiority. Specifically, the baseline methods include: (1) SDEdit Meng et al. (2021), which directly upsamples the guidance image, adds noise, and then uses a diffusion model to denoise it. Given its simplicity, SDEdit has been used in popular open-source projects [1] for high-resolution enhancement. (2) Skip-Residual, introduced in DemoFusion Du et al. (2024), constructs a complete progressive diffusion process and then injects low-resolution guidance at each timestep during denoising, thereby enhancing the guidance's effectiveness. (3) Time-aware scaled RoPE Zhuo et al. (2024), achieving the same idea as Relay Diffusion Teng et al. (2023) for butter efficiency but in a training-free manner by applying time-conditioned re-scaling to RoPE during the diffusion process.

In Fig. 8, we visualize the results of comparison. We can observe that, leveraging Projected Flow, I-Max significantly enhances the details of generated images in the 1K→4K extrapolation. While SDEdit also brings finer details, it lacks the ability to guide the flow direction throughout the entire denoising process, leading to the appearance of significant artifacts. In contrast, Skip-residual achieves better stability by injecting guidance at each timestep, but the intermediate guidance constructed by a diffusion process introduces fixed noise that cannot be adjusted based on the current latent representation, thus reducing image quality. Lastly, while Time-Aware Scaled RoPE offers superior efficiency, it fails to produce usable results during the $16\times$ extrapolation due to the absence of explicit high-quality guidance. Additionally, in Fig. 7, we also conduct GPT-4o preference evaluations between the three baseline methods and Projected Flow. The results were consistent with the visual findings, where Projected Flow achieved significantly higher than $50\%$ win rates.

---

[1] https://github.com/AUTOMATIC1111/stable-diffusion-webui/

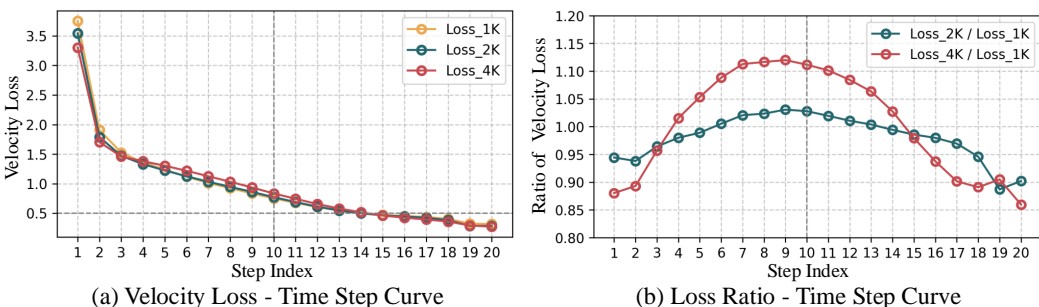

Figure 9: (a) Illustration of velocity loss values across different resolutions over the denoising process. (b) Illustration of the ratio of extrapolated-resolution to native-resolution velocity loss over the denoising process.

### 3.5 Why Do We Need Low-resolution Guidance?

In resolution extrapolation, using native-resolution generation as guidance is a common practice in existing works Du et al. (2024); Teng et al. (2023); Hwang et al. (2024); Lin et al. (2024a;b). In this paper, we also regard it as a core perspective for addressing resolution extrapolation and introduce Projected Flow as one of our primary contributions. In this section, we aim to demonstrate why low-resolution guidance is so critical and how, with the support of low-resolution guidance, resolution extrapolation can lead to the emergence of finer details and improvements in overall quality. Here, we conduct experiments using the 1K Lumina-Next model and 1,000 randomly sampled high-resolution text-image pairs. Specifically, we resize the images to a particular resolution, add noise up to a specific timestep, and then input the noisy images into the model to calculate the velocity loss. In this way, we can evaluate the model's performance at the given timestep. As shown in Fig. 9 (a), we plotted the loss/timestep curves for both native resolution (1K) and extrapolated resolution (2K and 4K). We can observe that as time $t$ progresses, all curves of different resolutions decrease, which is reasonable since the higher the signal-to-noise ratio, the easier it becomes to predict the velocity.

Furthermore, we compute the ratio of the loss at extrapolated resolutions to the loss at the native resolution. In that case, we can visualize the degree of the model's performance degradation at extrapolated resolutions. According to the results shown in Fig. 9 (b), we can observe that the most severe performance degradation occurs in the intermediate timesteps. Previous work Choi et al. (2022) has demonstrated that the core content generation of diffusion models happens during the middle stage of the denoising process, while the later stage is about high-frequency details refinement. This indicates that when generating at extrapolated resolutions, the model's ability to generate main content degrades noticeably, while the ability to refine local details is less resolution-sensitive. This finding clarifies the motivation for implementing low-resolution guidance – low-resolution guidance can alleviate the degradation of the model's main content generation ability as resolution increases, allowing the model to focus on refining local details, which is less sensitive to resolution changes.

Note that we observe a relatively small performance gap in the early stages of the denoising process. This is because when the signal-to-noise ratio is very low, there are many possible correct flow directions, and the model learns to predict the mean of data distribution. Therefore, the loss calculated using the per-sample velocity ground truth may not accurately reflect the model's performance. Additionally, we notice that the ratio in Fig. 9 (b) can sometimes be smaller than 1. This occurs because, as the sequence length increases, the scale of the RFT's predictions can change. It does not imply that the model performs better at higher resolutions. Instead of focusing on the numerical values of the curve, the overall trend of the curve offers more valuable insight.

### 3.6 Time Costs from Extrapolation.

In this paper, we focus on optimizing the performance of resolution extrapolation to approach the generative resolution limits of pre-trained RFTs, without considering acceleration techniques such

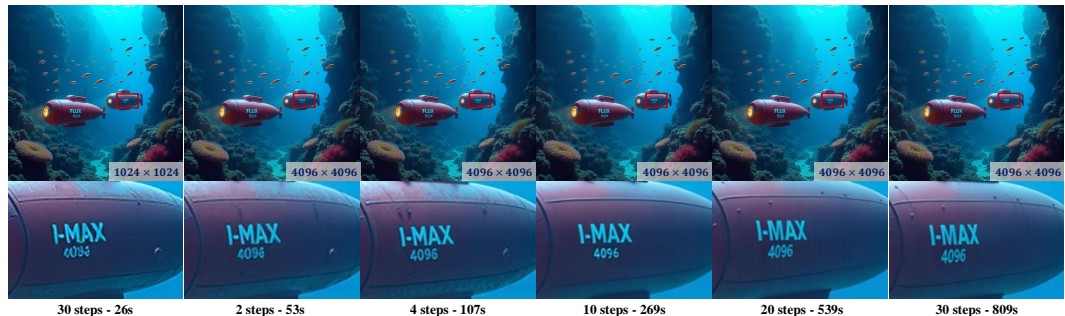

Figure 10: Illustration of I-Max results and inference costs at different inference step numbers.

as token merging Bolya et al. (2022) or deep cache Ma et al. (2024b). However, it is important to note that as resolution extrapolation increases, inference costs rise exponentially. Therefore, we analyze the generation quality and inference time across different inference steps in Fig. 10. Note that the 2-step and 4-step results are obtained using the Flux.1-schnell model, while the other results were produced using the Flux.1-dev model. We report the inference time of the denoising process, which is evaluated on a single A100 GPU. Based on the experimental results, we find that with the guidance of low-resolution generation results, even with very few inference steps (*e.g.*, 2 steps), I-Max can achieve better results at extrapolated resolutions than native resolution. Additionally, we observe a consistent improvement in the quality of generated details as the number of inference steps increased. This suggests a significant trade-off space between performance and efficiency, allowing for flexible adjustments according to the practical applications.

## 4 LIMITATION

In this paper, we propose I-Max to maximize the resolution potential of text-to-image models, *i.e.*, enabling them to generate high-quality images far beyond their training resolution. However, we found that the model's ability to extrapolate is significantly influenced by the foundation model – both the architecture and training strategy determine their behavior and capability during resolution extrapolation. For example, through sequential ablation studies, we observed that the DiT-structured Lumina-Next-2K and the MMDiT-structured Flux.1-dev exhibit different collapse modes. Moreover, while they both have the capability to generate at 2K resolution, Flux.1-dev is trained to accommodate a range from $256^2$ to $2048^2$ resolutions, which limits us to generate up to 4K with the help of I-Max. Nevertheless, we observe an overall trend where generation quality first improves and then degrades across different models (refer to Sec. 3.3), proving that I-Max has general applicability to RFTs.

Besides, in this work, we particularly focus on maximizing the generation quality during resolution extrapolation and do not address the issue of exponentially increasing inference costs associated with high-resolution generation. We experimented with some naive solutions, such as token merging or token dropping, but they led to noticeable performance degradation in the tuning-free setting. We believe this is an area worth exploring in future research.

## 5 CONCLUSION

In this paper, we propose a resolution extrapolation framework for rectified flow transformers, called I-Max. I-Max consists of two key components: (i) a newly proposed projected flow strategy that leverages the simplicity of rectified flow to implement low-resolution guidance, significantly reducing the inference complexity at extrapolated resolutions; (ii) an inference toolkit that ensures the model's generalization capability to extrapolated resolutions. Compared to previous approaches, I-Max significantly enhances the stability of the model when generating at extrapolated resolutions, which is why we claim it maximizes the resolution potential of text-to-image models. Through experiments, we demonstrate the effectiveness of the proposed method and the necessity of each design component.

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

# A APPENDIX

## A.1 GPT-4O PREFERENCE EVALUATION

# System Content
You are an evaluator specialized in image quality analysis for high-resolution text-to-image generation models.

# User Content
Please evaluate the following two generated images in terms of clarity, richness of detail, and overall quality. The prompt is: *A quirky miniature scene, potato chip soldiers parachuting onto a ceramic bowl filled with ridged potato chips, tiny plastic figurines suspended by yellow mushroom cloud-like parachutes, surreal food photography, soft lighting*. Please do not be affected by the order of the images. Output [Image 1] if the first image is better, [Image 2] if the second image is better, and give me the reason.

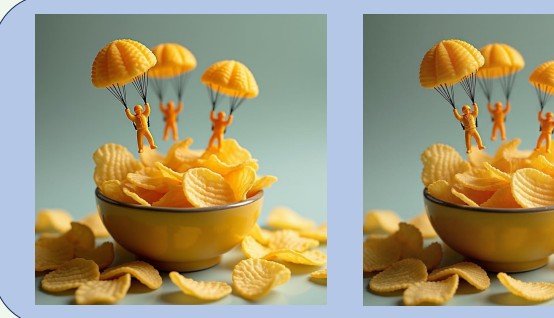

[Image 2]
The second image has a sharp colour palette, with details of the skydiver and chips clearly visible, and an even light and shadow treatment that provides a better visual effect.

Figure 11: Illustration of GPT-4o preference evaluation.

In this paper, we follow Pixart-$\Sigma$ to use GPT-4o for preference evaluation. In Fig. 11, we present a use case of such an evaluation. During practical implementation, we found that GPT-4o's judgments generally align with human preferences and are independent of irrelevant information, such as the order of the images. Additionally, we prompt GPT-4o to provide reasoning for its judgments, encouraging it to think carefully and allowing us to assess the quality of the evaluation.

## A.2 SEQUENTIAL ABLATION ON LUMINA-NEXT-2K

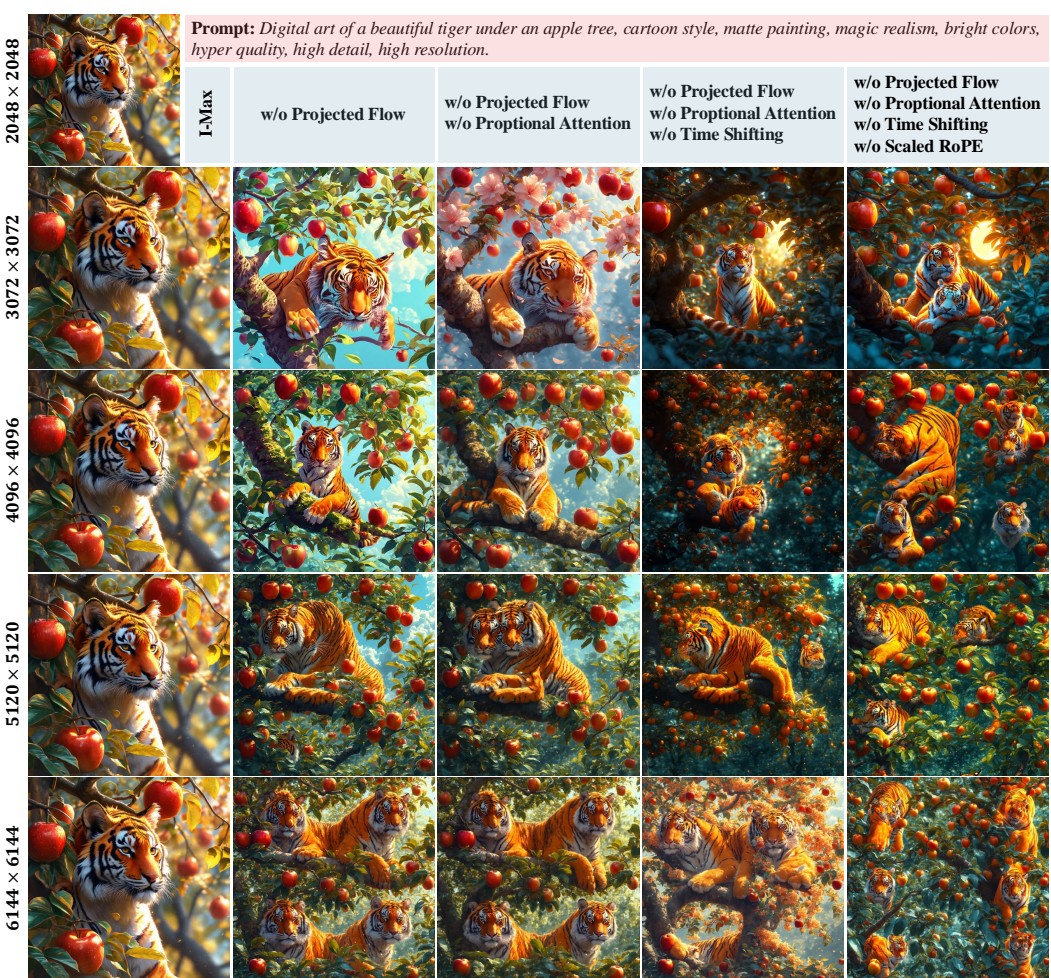

Figure 12: Sequential ablation of I-Max. We illustrate the effect of sequentially removing different components of I-Max on Lumina-Next-2K across the 2K→6K resolution range.

Here, we provide the results of sequential ablation on Lumina-Next-2K. Unlike Flux.1-dev, which is based on the MMDiT architecture, Lumina-Next-2K uses cross-attention blocks to inject text information. This structural difference leads to distinct failure modes during resolution extrapolation between the two base models, with Lumina-Next-2K tending to generate repetitive patterns. However, we still arrive at a consistent conclusion – every component of I-Max contributes significantly and positively during the resolution extrapolation process.

## A.3   COMPARISON WITH SUPER-RESOLUTION

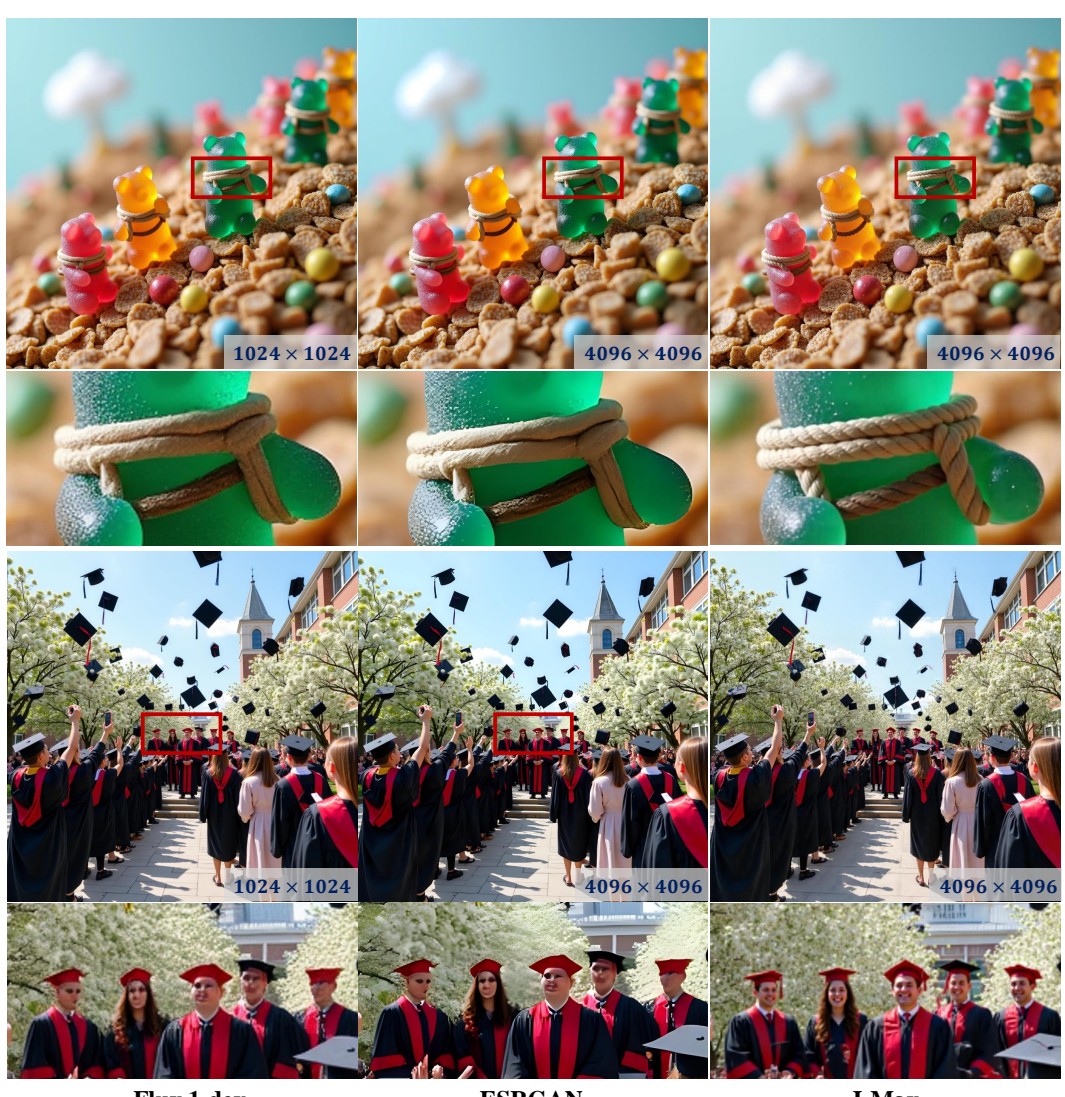

**Flux.1-dev**    **ESRGAN**    **I-Max**

Figure 13: Comparison with super-resolution method ESRGAN.

For I-Max, a potential concern is that we could synthesise high-resolution images by using a pipeline of low-resolution generation followed by super-resolution, which offers higher inference efficiency than resolution extrapolation. This is particularly relevant since I-Max itself uses low-resolution generated results as guidance. Therefore, we would like to clarify that resolution extrapolation serves a different purpose from super-resolution. Specifically, high-resolution generation focuses on producing high-quality, highly detailed images, while super-resolution has a strong requirement for maintaining consistency between input and output, which often limits the detail enhancement in the output. In Fig. 13, we also compare I-Max with the commonly used super-resolution method ESRGAN Wang et al. (2018). It is evident that the rich details introduced by I-Max via high-resolution generation are not present in super-resolution results. Moreover, ESRGAN cannot correct artifacts present in the generated images.

## A.4 ANY RESOLUTION AND ANY ASPECT-RATIO GENERATION

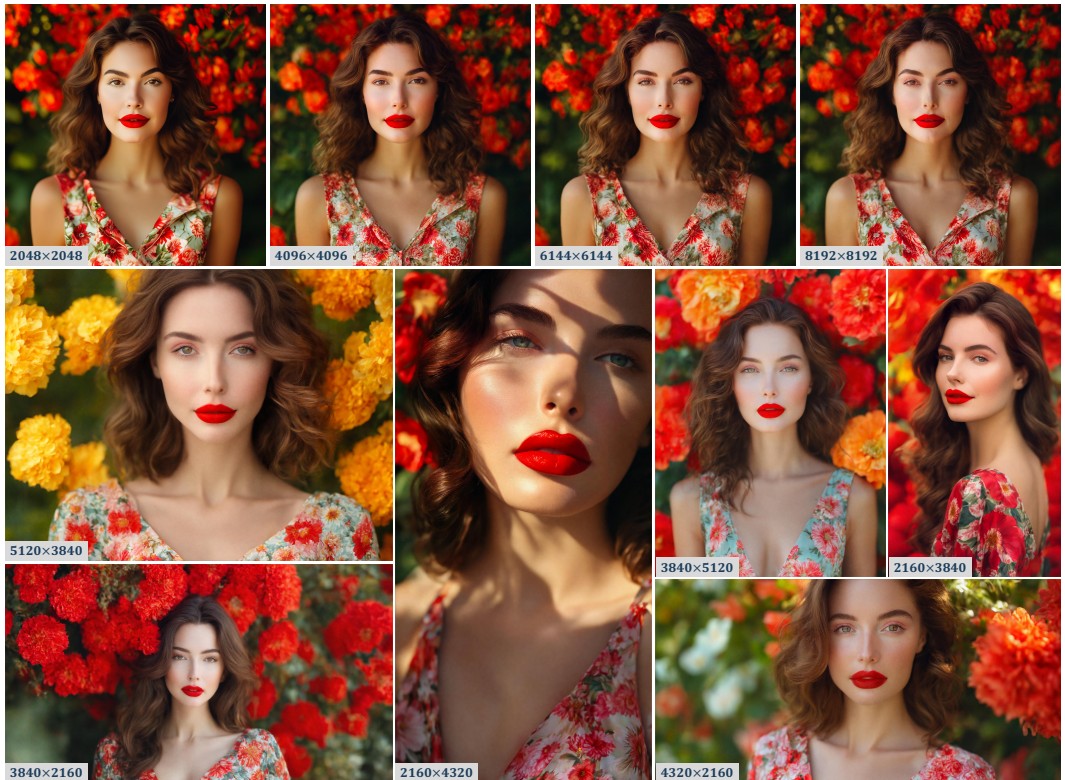

Figure 14: Generation results with various resolutions and aspect-ratios.

In Fig. 14, we provide images generated by Lumina-Next-2K with I-Max at arbitrary resolutions and aspect ratios using the same prompt: "*A close-up portrait of a young woman with flawless skin, wavy brown hair, red lipstick, wearing a vintage floral dress and standing in front of a blooming garde, ndetailed, vivid color, 8k*". This illustrates that I-Max can be stably applied to any image size.

