# OpenReview forum: "I-Max: Maximize the Resolution Potential of Pre-trained Rectified Flow Transformers with Projected Flow"
_ICLR.cc/2025/Conference — Submitted to ICLR 2025_

### Official Review · Reviewer_eZ4Q · 2024-11-01

**Soundness:** 2
**Presentation:** 2
**Contribution:** 3
**Rating:** 5
**Confidence:** 3

**Summary:**

This paper proposes the I-Max, designed to maximize the resolution potential of Text-to-Image Rectified Flow Transformers (RFTs). I-Max includes a novel Projected Flow strategy and an advanced inference toolkit, enhancing generative stability, improving image detail, and correcting artifacts during resolution extrapolation.

**Strengths:**

- The paper is well-structured.
- The proposed method achieves excellent visual results.

**Weaknesses:**

1. **Lack of Quantitative Comparisons:**
   The paper lacks any quantitative comparisons, making it difficult to demonstrate the superiority of the proposed method. Metrics such as FID (Fréchet Inception Distance) and IS (Inception Score) could be used to provide concrete quantitative comparisons.

2. **Need for User Study:**
   A user study is necessary to validate the visual effectiveness of the method. This study should focus on aspects such as detail preservation, artifact reduction, and overall image quality of the generated images, which would further enhance the quality of the paper.

3. **Comparison of Model Parameters and Runtime:**
   The paper should include comparisons of model parameters and runtime to provide a comprehensive picture of the method’s efficiency. Reporting the generation time at different resolutions, such as 1K and 2K, is crucial for understanding the practical applicability and efficiency of the proposed framework.

**Questions:**

As shown in Weaknesses

---

> ### Author Response · Authors · 2024-11-18
>
> **W1. Lack of Quantitative Comparisons.**
>
> Thanks! Relying solely on a single metric indeed weakens the persuasiveness of the paper. Referring to widely accepted works on high-resolution synthesis [1,2], we use 1,000 prompts randomly sampled from LAION-5B as the test set, along with 10,000 randomly sampled images as the reference set, to calculate FID, FID_crop, and CLIP Score. This allows us to comprehensively evaluate both the global and local image quality, as well as the text-image consistency.
>
> In the table below, we report the evaluation results of Flux.1-dev + I-Max at different resolutions. We can observe that the results were highly consistent with those from the GPT-4o preference evaluation. Both two image quality metrics (FID and FID_crop) show improvement as the resolution increased, achieving optimal performance at 3K resolution, echoing the results in Fig. 6. For the text-image consistency metric, CLIP Score, we observed a very slight decrease as the resolution increased, which is understandable since I-Max guides high-resolution generation using the output at 1024×1024 resolution. Thus, the text-image consistency is not further enhanced.
>
> | Different Resolution | FID | FID_crop | Clip Score |
> | --- | --- | --- | --- |
> | 1024×1024 | 63.47 | 63.95 | 28.52 |
> | 2048×2048 | 57.69 | 55.59 | 28.49 |
> | 3072x3072 | 53.21 | 52.91 | 28.46 |
> | 4096x4096 | 53.75 | 53.01 | 28.41 |
>
> In addition, the Black Forest Lab Team claims that Flux.1-dev can directly generate images at 2048×2048, but it has been noticed that the generation quality at 2048×2048 is not particularly good enough. Therefore, we also compared the results of its default settings against the results with I-Max at the 2048×2048 resolution. I-Max also achieves a significant margin of improvement, indicating its practical value in real-world applications.
>
> | Different Step Numbers | FID | FID_crop | Clip Score |
> | --- | --- | --- | --- |
> | Default Setting | 59.10 | 75.53 | 28.29 |
> | Enhanced with I-Max | 53.75 | 53.01 | 28.41 |
>
> As for the comparison between different low-resolution guidance approaches, we report the results in the table below. All metrics show strong consistency with the GPT-4o preference evaluation in Fig. 7, demonstrating the superiority of the proposed projected flow approach.
>
> | Different Guidance | FID | FID_crop | Clip Score |
> | --- | --- | --- | --- |
> | Projected Flow | 53.75 | 53.01 | 28.41 |
> | SDEdit | 54.89 | 55.76 | 28.32 |
> | Skip Residual | 53.94 | 53.89 | 28.40 |
> | Time-aware Scaled RoPE | 54.93 | 54.37 | 27.96 |
>
> We will revise the paper and include these results.
>
> **W2. Need for User Study.**
>
> Good advice! Human preference evaluation is indeed the most direct method for assessing the performance of generative models. However, during the rebuttal phase, it is challenging to organize a sufficiently large-scale user study. Instead, we utilized GPT-4o to simulate preference evaluation in the paper. As demonstrated in existing work [3], GPT-4o preferences have been validated to align closely with human preferences. In the camera-ready version, we will also supplement the quantitative results derived during the rebuttal phase to make the paper more convincing.
>
> **W3. Comparison of Model Parameters and Runtime.**
>
> Well said! As a training-free resolution extrapolation algorithm, I-Max does not introduce any additional model parameters. Regarding efficiency, the table below presents the inference time costs at different resolutions under varying numbers of inference steps.
>
> | Different Resolution | 4 Steps | 10 Steps | 30 Steps |
> | --- | --- | --- | --- |
> | 1024×1024 | 3.4s | 8.6s | 26s |
> | 2048×2048 | 33s | 48s | 147s |
> | 3072x3072 | 54s | 111s | 301s |
> | 4096x4096 | 107s | 539s | 809s |
>
> Additionally, we provide quantitative evaluation results at 4096×4096 resolution for different inference step numbers to demonstrate the trade-off between inference cost and generation quality in I-Max. What we observe is that, thanks to the extrapolation stability brought by projected flow, the quality of generated results shows only a slight degradation as the number of inference steps decreases, effectively reducing the high computational costs associated with high-resolution inference. Similar conclusions can also be drawn from the qualitative results shown in the Fig. 10 of our submission.
>
> | Different Step Numbers | FID | FID_crop | Clip Score |
> | --- | --- | --- | --- |
> | 4 Steps | 54.63 | 58.10 | 28.35 |
> | 10 Steps | 53.99 | 53.72 | 28.39 |
> | 30 Steps | 53.75 | 53.01 | 28.41 |
>
> We will revise the paper and include these results.
>
> *[1] Du, Ruoyi, et al. "Demofusion: Democratizing high-resolution image generation with no $$$", CVPR, 2024.*
>
> *[2] Yang, Zhuoyi, et al. "Inf-dit: Upsampling any-resolution image with memory-efficient diffusion transformer", ECCV, 2024.*
>
> *[3] Chen, Junsong, et al. "Pixart-\sigma: Weak-to-strong training of diffusion transformer for 4k text-to-image generation." arXiv preprint arXiv:2403.04692 (2024).*

---

> > ### Comment · Reviewer_eZ4Q · 2024-11-24
> >
> > Thank you for your response. However, my concerns have not been fully addressed.
> >
> > 1.	Lack of Quantitative Comparisons: I was expecting to see a comparison of your proposed method against existing state-of-the-art methods to substantiate its superiority. Simply showcasing the performance of your method at different resolutions does not convincingly demonstrate its advantages. It is essential to include a quantitative comparison with other established methods to validate the performance claims of your work.
> >
> > 2.	Comparison of Model Parameters and Runtime: Similarly, the evaluation should include a comparison of the performance and computational complexity of your method with existing approaches. Presenting only the metrics of your method does not provide a clear indication of its superiority. It is crucial to compare model parameters and runtime with other state-of-the-art methods to highlight any potential benefits.
> >
> > There are numerous existing works on arbitrary-scale/high-resolution image generation, including but not limited to:
> >
> > [1] Du, Ruoyi, et al. “Demofusion: Democratizing high-resolution image generation with no $$$”, CVPR, 2024.
> >
> > [2] Yang, Zhuoyi, et al. “Inf-dit: Upsampling any-resolution image with memory-efficient diffusion transformer”, ECCV, 2024.
> >
> > [3] Chen, Junsong, et al. “Pixart-σ: Weak-to-strong training of diffusion transformer for 4k text-to-image generation.” arXiv preprint arXiv:2403.04692 (2024).
> >
> > [4] A frequency perspective on training-free high-resolution image synthesis.
> >
> > [5] Scalecrafter: Tuning-free higher-resolution visual generation with diffusion models.
> >
> > [6] UltraPixel: Advancing Ultra-High-Resolution Image Synthesis to New Peaks.
> >
> > I believe the lack of comparative analysis with these existing methods is the primary reason I am unable to adjust my score. A comprehensive comparison would significantly strengthen the justification of your method’s effectiveness and efficiency.

---

> > > ### Author Response · Authors · 2024-12-02
> > > **Looking forward to your response**
> > >
> > > Dear Reviewer eZ4Q,
> > >
> > > Thank you for your feedback in the previous round!
> > >
> > > Considering the discussion period is about to end, we sincerely look forward to your replies. We hope that we have addressed your concerns regarding "Lack of Quantitative Comparisons"; if not, we are happy to listen to the follow-up questions. Thanks!
> > >
> > > Bests,
> > >
> > > Authors of Submission 2180

---

> ### Author Response · Authors · 2024-11-25
>
> Thanks. I understand that the reviewer expects us to compare I-Max with SOTA high-resolution generation methods in terms of performance, parameter count, and runtime. However, among the mentioned methods, Inf-DiT [2] is a super-resolution model, PixArt-Sigma [3] and UltraPixel [6] are tuning-based approaches, while the remaining methods — DemoFusion [1], ScaleCrafter [5], and FouriScale [4] — focus on resolution extrapolation (i.e., training-free high-resolution generation) and share the same motivation as I-Max. In the existing literature, it is uncommon to directly compare tuning-based methods with training-free methods, considering the advantage of tuning-free methods lies in the absence of tuning costs.. Therefore, we will prioritize discussing these training-free competitors [1,4,5].
>
> 1. **Comparison of performance.**
>
> Factually, we conducted comparisons with SOTA training-free methods in the paper, albeit in a reimplementation manner. Considering that I-Max is the first approach tailored for rectified flow transformers, whereas previous works are primarily applied to UNet-based LDMs (e.g., SDXL), to avoid unfair comparisons, we first need to reimplement these methods on the same base model with the same inference techniques — we all know that Flux.1-dev significantly surpasses SDXL. In the table below, our comparative experiments on different guidance methods essentially include: (1) reimplementation of SOTA methods on Flux.1-dev, e.g., Skip-residual from DemoFusion; (2) advanced methods under the DiT architecture, e.g., Time-aware Scaled RoPE from Lumina-Next; (3) widely used model-agnostic method, e.g., SDEdit.
>
> | Different Guidance | FID | FID_crop | Clip Score |
> | --- | --- | --- | --- |
> | Projected Flow (I-Max) | 53.75 | 53.01 | 28.41 |
> | SDEdit [7] | 54.89 | 55.76 | 28.32 |
> | Skip Residual (DemoFusion [1]) | 53.94 | 53.89 | 28.40 |
> | Time-aware Scaled RoPE (Lumina-Next [8]) | 54.93 | 54.37 | 27.96 |
>
> ScaleCrafter [5] and its variants (e.g., FouriScale [4]) rely on dilated convolution for resolution extrapolation, which is specifically tailored for UNet-based LDMs and not suitable for DiT-based LDMs. This further highlights the unique value of I-Max's exploration.
>
> 2. **Comparison of parameter count and runtime.**
>
> The training-free nature of resolution extrapolation approaches ensures that they do not introduce additional parameters, meaning the model's parameter count is entirely determined by the base model used. As for runtime, here we provide a cross-model comparison of inference times between I-Max and the mentioned methods. Thanks to the stability of I-Max (please refer to our previous response), it requires fewer inference steps, achieving comparable inference times to methods based on 3.5B models while utilizing a 12B model. All the results are tested on a single A100 GPU.
>
> |  | ScaleCrafter [5] | FouriScale [4] | DemoFusion [1] | I-Max (10 Steps) | I-Mqx (30 Steps) |
> | --- | --- | --- | --- | --- | --- |
> | Base Model | SDXL | SDXL | SDXL | Flux.1-dev | Flux.1-dev |
> | Parameter | 3.5B | 3.5B | 3.5B | 12B | 12B |
> | Runtime (2048×2048) | 41s | 73s | 94s | 48s | 147s |
> | Runtime (4096x4096) | 585s | 563s | 728s | 539s | 809s |
>
> 3. **Summary**
>
> Overall, I understand that a cross-model comparison can show how I-Max compares with other T2I Pipelines [1,2,3,4,5,6] in terms of effectiveness. However, as a research paper, I would rather divide the problem into several key techniques and provide useful insights to the community through fair experiments. In addition, resolution extrapolation focuses on unlocking the potential of pre-trained LDMs. Therefore, we argue that the vertical comparison in the paper are of particular value. Compared with existing works [1,4,5], we are the first to achieve that the model's overall performance at the extrapolated resolution can surpass that at the native resolution (e.g., the FID at 4096 resolution is better than the FID at 1024 resolution).
>
> [1] Du, Ruoyi, et al. “Demofusion: Democratizing high-resolution image generation with no $$$”, CVPR, 2024.
>
> [2] Yang, Zhuoyi, et al. “Inf-dit: Upsampling any-resolution image with memory-efficient diffusion transformer”, ECCV, 2024.
>
> [3] Chen, Junsong, et al. “Pixart-σ: Weak-to-strong training of diffusion transformer for 4k text-to-image generation.” arXiv preprint arXiv:2403.04692 (2024).
>
> [4] A frequency perspective on training-free high-resolution image synthesis.
>
> [5] Scalecrafter: Tuning-free higher-resolution visual generation with diffusion models.
>
> [6] UltraPixel: Advancing Ultra-High-Resolution Image Synthesis to New Peaks.
>
> [7] Meng, Chenlin, et al. "SDEdit: Guided Image Synthesis and Editing with Stochastic Differential Equations." *ICLR*.
>
> [8] Zhuo, Le, et al. "Lumina-next: Making lumina-t2x stronger and faster with next-dit." NeurIPS, 2024.

---

### Official Review · Reviewer_Jtf5 · 2024-11-03

**Soundness:** 3
**Presentation:** 2
**Contribution:** 3
**Rating:** 6
**Confidence:** 4

**Summary:**

This paper presents a method to extrapolate the resolution of the generated images at inference time. The authors focus on rectified flow transformers. The key ideas are a projected flow strategy that is designed to ensure more stability at inference, and a number of implementation techniques to enhance the quality of the extrapolation, such as NTK-aware scaled RoPE, SNR resolution adjustment, attention re-scaling, and text duplication.

**Strengths:**

I-Max integrates a number of simple but important components to make a rectified flow model generalise to higher resolutions at inference. In particular, the projected flow strategy makes sense as a method to ensure more stability. As far as I know this is original and the specific implementation in the style of a classifier-free guidance seems original too.
The results achieved in the experimental section show also that the proposed projection with the other inference techniques are quite effective in the resolution extrapolation task.

**Weaknesses:**

The presentation is at times not optimal.
For example, the split in the introduction into How to guide and How to infer does not seem very clear to me. At lines 93-95 the explanations do not seem to match the names of the two perspectives.
Overall, the use of the English language could be better. I would suggest to have the paper revised but a native English speaker to correct typos.
Could you check the following?
Line 220: Eq. 2 illustrates the equivariance of the flow wrt the projection rather than its invariance.

The other concern is regarding the method (see also the Questions below). It would be useful to the reader to better explain the technical choices by providing the motivation/rationale behind each of them.

**Questions:**

I would like the authors to clarify the following points:
1) Provide visual examples of failures at very high resolution (as pointed out in sec 3.3 for Figure 6);
2) Why is the projected flow implemented via the classifier-free guidance? Is this the only way?
3) Could you show how you would explain the transition to eq. (5) with more technical details?

---

> ### Author Response · Authors · 2024-11-18
>
> **W1. The presentation is at times not optimal.**
>
> Thanks for the advice!
>
> - Regarding Lines 93-95, we revisit existing work from the perspectives of "How to guide?" and "How to infer?" because they primarily address the problem from two angles: (i) how to use low-resolution generation results to provide content guidance for high-resolution generation, and (ii) how to enhance the model's generalization ability for high-resolution inputs during the inference phase. Perhaps we could use more straightforward terms to distinguish these two aspects, such as "low-resolution guidance" and "extrapolated-resolution generalization."
> - For Line 220, we greatly appreciate the reviewer pointing this out! What we intended to express was **equivariance**, not **invariance**. We will correct this oversight in the camera-ready version.
>
> Currently, the presentation of this paper is admittedly less than perfect, and we will try to polish it as comprehensively as possible.
>
> **Q1. Provide visual examples of failures.**
>
> Good point! We should provide failure cases to qualitatively demonstrate the model's behaviour when approaching the limits of its capabilities. A common failure mode can be understood as the generation of highly grainy images with unexpected high-frequency textures when the model's prior knowledge fails to support extrapolation to higher resolutions. Considering that we cannot include images during the rebuttal phase, we will supplement the corresponding results and discussions in the camera-ready version.
>
> **Q2. Why is the projected flow implemented via the classifier-free guidance?**
>
> Apologies for the confusion. When we mentioned classifier-free guidance, we did not intend to imply a theoretical connection between projected flow and classifier-free guidance — we simply aimed to convey that projected flow is also implemented as a guidance term. We will revise the paper to clarify this point. As for whether this guidance can be implemented in other forms, we have supplemented some experimental results here to demonstrate the necessity of certain design choices.
>
> - A guidance scale that decreases over time t is not a conventional approach. In the table below, we compare the cosine decay guidance scale (from 1 to 0) with two constant guidance scales (0.5 and 1.0). All three metrics (FID, FID_crop, CLIP Score) indicate the necessity of the cosine decay strategy, as it aligns with the working characteristics of diffusion models — restoring low-frequency structures in the early stages and high-frequency details in the later stages.
>
>
>     | Guidance Strategy | FID | FID_crop | Clip Score |
>     | --- | --- | --- | --- |
>     | Cosine Decay | 53.75 | 53.01 | 28.41 |
>     | Constant 0.5 | 54.98 | 54.32 | 28.22 |
>     | Constant 1.0 | 55.67 | 55.41 | 27.76 |
> - The projection function we use is also customizable. In the paper, we by default use low-pass filtering based on Discrete Wavelet Transform. In the table below, we report the evaluation results for three types of filters: (1) Discrete Wavelet Transform, (2) Fast Fourier Transform, and (3) low-pass filtering based on downsampling-upsampling. We observe that different filters do not have a significant impact on the performance of I-Max, with DWT showing slightly better overall performance.
>
>
>     | Different Projection | FID | FID_crop | Clip Score |
>     | --- | --- | --- | --- |
>     | DWT | 53.75 | 53.01 | 28.41 |
>     | FFT | 53.64 | 53.56 | 28.37 |
>     | Downsample-Upsample | 53.98 | 53.57 | 28.43 |
>
> We will include these results and the corresponding discussion in the camera-ready version.
>
> **Q3. Technical details about the transition to Eq. (5).**
>
> If the reviewer's question is about how Eq. (4) leads to Eq. (5), the theoretical details behind it are actually quite straightforward. Let us consider a simpler setting. When constructing a rectified flow model at a fixed resolution, we have:
>
> $$
> dX_t = (X_1 - X_0) dt.
> $$
>
> Similarly, we can express $X_t$ as:
>
> $$
> X_t = X_0 + dX_t \cdot dt.
> $$
>
> Then it follows that:
>
> $$
> dX_t = \frac{X_1 - X_t}{1 - t} dt.
> $$
>
> However, in this context, $X_t$ represents the intermediate state at time $t$ , while $dX_t$ denotes the increment, which might cause confusion. To prevent ambiguity, we can update the notation to:
>
> $$
> dX_t = \frac{X_1 - X_\tau}{1 - \tau} dt.
> $$
>
> Similarly,  Eq. (5) can be rewritten as:
>
> $$
> dP(X_t^\text{extra}) \approx \frac{\hat{X_1}^\text{native} - P(X_\tau^\text{extra})}{1-\tau} dt.
> $$
>
> We'll modify all the corresponding notations in the camera-ready version.

---

> > ### Comment · Reviewer_Jtf5 · 2024-11-25
> >
> > Dear authors, thanks for your answers.

---

### Official Review · Reviewer_Xw7y · 2024-11-04

**Soundness:** 3
**Presentation:** 2
**Contribution:** 2
**Rating:** 5
**Confidence:** 4

**Summary:**

The paper addresses the task of tuning-free resolution extrapolation for text-to-image Rectified Flow Transformers (RFTs) from which one can obtain samples at a much higher resolution than the resolution at which the model was originally trained. While directly training high-resolution generative models is practically difficult, this paper aims to adapt trained RFTs to generate images of high resolutions (such as 4096X4096) without the need for fine tuning.

The proposed scheme named I-Max involves low-resolution guidance named projected flow. Here, the low-resolution
space is treated as a low-dimensional projection of the high-resolution space, and thereby the low-resolution
flow can be regarded as the projection of the ideal high-resolution flow. Considering the linear
interpolation characteristic of rectified flow, I-Max incorporates guidance in the projected space at each timestep.
Additionally I-max incorporates inference techniques tailored for RFT to enhance the model’s ability to generalize to extrapolated resolution.

**Strengths:**

Attempting to extrapolate resolution of trained models is interesting and practically useful given the issues of data quality and fine tuning costs. This paper addresses the task of resolution extrapolation for trained Rectified Flow Transformers for the first time.

Relevant prior works have been discussed appropriately.

**Weaknesses:**

The guidance mechanism in eq 7 does not exactly correspond to the Classifier Free Guidance. It is indeed some sort of a guidance function. Could the authors explain the relationship between their guidance mechanism and Classifier Free Guidance?

 Additionally, it is not clear how the first term in the RHS of eqns 6 and 7 (v_{theta} at the extra resolution) is obtained. Is the same model trained at the native resolution used for this?


The steps followed to generate the high resolution image could have been summarized in the form of an algorithm.



Evaluation is based only on GPT-4o. More qualitative examples wherein one can see improvements as shown in Fig 2 could have been shown in the supplementary material. Since GPT-4o is not necessarily trained for image quality assessment, other measures should have been used for comparison. Time aware scaled ROPE (Fig 7) also has good performance according to this measure.

Typo 'for butter efficiency' line 419.


Some of the ideas incorporated are based on existing works. Specifically, the inference techniques in section 2.3 are based on prior works.
Could the authors clarify their novel contributions in the techniques used in section 2.3?

**Questions:**

Why is the proposed method not shown on only Lumina-Next instead of the self-trained Lumina-Next-2K? Does the method require native resolution also to be high? Can the proposed I-Max work for low native resolutions?

---

> ### Author Response · Authors · 2024-11-18
>
> **W1. The relationship with Classifier-free Guidance.**
>
> Sorry for the confusion. The guidance term in projected flow does not have a direct theoretical relationship to classifier-free guidance. We agree with the reviewer's point that this is simply a guidance function, and our mention of classifier-free guidance was intended to illustrate that projected flow is essentially a form of guidance. We will modify the statement that "in the form of classifier-free guidance" to "in a similar form of classifier-free guidance" to eliminate misunderstanding.
>
> **W2. How is term v_{theta} obtained?**
>
> Yes, Term v_{theta} is derived directly from inference using the model trained on native resolution applied to extrapolated resolution. To ensure the model achieves optimal generalization capability for extrapolated resolution, we incorporated several inference techniques, including scaled RoPE, time shifting, proportional attention, and text duplication, as illustrated in Fig. 4.
>
> **W3. Summarize the inference process in the form of an algorithm.**
>
> Thanks for the suggestion! In the revised version, we will present the complete inference process in the form of an algorithm to facilitate readers' understanding.
>
> **W4. Questions about evaluation.**
>
> Well said! Since additional images cannot be included during the rebuttal phase, we will supplement more qualitative results in the camera-ready version as per the reviewers' suggestions. Here, to further support the conclusions drawn from the GPT-4o preference evaluation, we supplement the evaluation results under the metrics of FID, FID_crop, and CLIP Score.
>
> In the table below, we report the evaluation results of Flux.1-dev + I-Max at different resolutions. We can observe that the results were highly consistent with those from the GPT-4o preference evaluation. Both two image quality metrics (FID and FID_crop) show improvement as the resolution increased, achieving optimal performance at 3K resolution, echoing the results in Fig. 6. For the text-image consistency metric, CLIP Score, we observed a very slight decrease as the resolution increased, which is understandable since I-Max guides high-resolution generation using the output at 1K resolution.
>
> | Different Resolution | FID | FID_crop | Clip Score |
> | --- | --- | --- | --- |
> | 1024×1024 | 63.47 | 63.95 | 28.52 |
> | 2048×2048 | 57.69 | 55.59 | 28.49 |
> | 3072x3072 | 53.21 | 52.91 | 28.46 |
> | 4096x4096 | 53.75 | 53.01 | 28.41 |
>
> As for the comparison between different low-resolution guidance approaches, we report the results in the table below. All metrics show strong consistency with the GPT-4o preference evaluation in Fig. 7, demonstrating the superiority of the proposed projected flow approach. It is worth noting that in Fig. 7, we present the win rate of projected flow compared to the baseline methods, where higher values indicate a more significant advantage of projected flow.
>
> | Different Guidance | FID | FID_crop | Clip Score |
> | --- | --- | --- | --- |
> | Projected Flow | 53.75 | 53.01 | 28.41 |
> | SDEdit | 54.89 | 55.76 | 28.32 |
> | Skip Residual | 53.94 | 53.89 | 28.40 |
> | Time-aware Scaled RoPE | 54.93 | 54.37 | 27.96 |
>
> Besides, we will also provide detailed analysis about the trade-off of inference costs and generation quality, please refer to our response to the Reviewer eZ4Q.
>
> **W5. Typo.**
>
> Thanks! We will correct the typo and carefully check other parts of the paper.
>
> **W6. Some of the ideas incorporated are based on existing works.**
>
> Good point! Some of the ideas in this paper were indeed proposed in previous literature, and we have carefully cited them. What we have done in I-Max is to revisit existing resolution extrapolation methods and, specifically for rectified flow transformers, integrate useful ideas (e.g., scaled RoPE and time-shifting) while proposing new techniques (e.g., projected flow and text duplication) to achieve performance that is reliable for practical applications. Without emphasizing SOTA performance, the newly introduced techniques are undoubtedly our contributions. Additionally, our analysis and validation of existing ideas on advanced base models also provide meaningful insights for the field, e.g., as demonstrated in Fig. 4 and Fig. 12, these techniques exhibit different behaviours across various model architectures (DiT vs MMDiT).
>
> We will further clarify our contributions in the revision.
>
> **Q1. Why not use Lumina-Next?**
>
> We use Lumina-Next-2K to validate that I-Max can be applied to: (i) different foundation models, (ii) different model architectures (MMDiT and standard DiT), and (iii) models with varying native resolutions (hence the fine-tuning of the model). In fact, a 2K fine-tuned SFT model allows us to generate 8K images through the same scale factor, demonstrating that the resolution extrapolation capability enabled by I-Max does not diminish as the native resolution increases. This makes it more valuable compared to models with lower native resolutions.

---

> > ### Comment · Reviewer_Xw7y · 2024-11-24
> >
> > Could the authors comment on these questions in my review
> > "Does the method require native resolution also to be high? Can the proposed I-Max work for low native resolutions?"
> > The authors could also mention the abilities of other resolution extrapolation methods on this aspect.

---

> > > ### Author Response · Authors · 2024-11-25
> > >
> > > Thank you for your response! Sorry that my previous reply wasn't straightforward enough. I-Max doesn't require the model to have an ultra-high native resolution. We are not using a 2K SFT model not to reduce the difficulty of resolution extrapolation. In the paper, we conduct the 1K -> 4K image generation of I-Max + Flux.1-dev and the 2K -> 8K image generation of I-Max + Lumina-Next-2K, and the degree of upscaling does not change. What we want to convey is that with the increase in the training resolution of the generation model, I-Max can consistently push up the upper limit of the generation resolution.

---

> > > ### Author Response · Authors · 2024-12-02
> > > **Looking forward to your response**
> > >
> > > Dear Reviewer Xw7y,
> > >
> > > Considering the discussion period is about to end, we sincerely look forward to your replies. We hope that we have addressed your concerns; if not, we are happy to listen to the follow-up questions. Thanks!
> > >
> > > Bests,
> > >
> > > Authors of Submission 2180

---

### Official Review · Reviewer_jzk2 · 2024-11-04

**Soundness:** 2
**Presentation:** 3
**Contribution:** 3
**Rating:** 5
**Confidence:** 3

**Summary:**

This paper introduces the I-Max framework, designed to address resolution extrapolation in text-to-image models using Rectified Flow Transformers. This paper incorporates a Projected Flow for fidelity and an inference toolkit to enhance model generalization at extrapolated resolutions. Experiments on the Lumina-Next-2K and Flux.1-dev models demonstrate that I-Max effectively improves the stability and detail of extrapolated high-resolution images, showing its potential for practical applications where tuning-free resolution extrapolation is needed.

**Strengths:**

- The paper is well-structured, making the methodology and findings easy to understand.
- The paper provides thorough experimental evaluations, including ablation studies on key components of the I-Max framework.

**Weaknesses:**

- The evaluation relies heavily on a single metric (GPT-4 preference) to assess the quality of generated images, limiting the objectivity of the results. This reliance may affect the demonstration of the proposed method’s effectiveness.
If evaluating high-resolution images with widely used metrics (e.g., FID) is challenging, as noted in the manuscript, a toy experiment using a pretrained model on a lower-resolution dataset, such as CIFAR or ImageNet, could offer a feasible benchmark and enable standardized metric comparisons.
- In line 254, the paper mentions the use of a low-pass filter for projection but does not specify the type. Additionally, exploring the impact of different low-pass filters could offer insights into how they affect stability and quality during resolution extrapolation.

**Questions:**

Please refer to the Weaknesses section.

---

> ### Author Response · Authors · 2024-11-18
>
> **W1. A single metric limits the objectivity of the results.**
>
> Thanks! Relying solely on a single metric indeed weakens the persuasiveness of the paper. Referring to widely accepted works on high-resolution synthesis [1,2], we found that a feasible approach to evaluate the image quality of high-resolution images is to consider both the FID metric for the entire images and for image crops (i.e., FID and FID_crop). Meanwhile, metrics for evaluating text-image consistency, such as CLIP Score, are not affected by resolution and can be applied directly to high-resolution images. Therefore, in this rebuttal, we follow the approach in [1], using 1,000 prompts randomly sampled from LAION-5B as the test set, along with 10,000 randomly sampled images as the reference set, to calculate FID, FID_crop, and CLIP Score. This allows us to comprehensively evaluate both the global and local image quality, as well as the text-image consistency.
>
> In the table below, we report the evaluation results of I-Max at four different resolutions (1024×1024, 2048×2048, 3072×3072, and 4096×4096) when using Flux.1-dev as the foundation model. We can observe that the results were highly consistent with those from the GPT-4o preference evaluation. Both two image quality metrics (FID and FID_crop) show improvement as the resolution increased, achieving optimal performance at 3K resolution, echoing the results in Fig. 6. For the text-image consistency metric, CLIP Score, we observed a very slight decrease as the resolution increased, which is understandable since I-Max guides high-resolution generation using the output at 1024×1024 resolution. Thus, the text-image consistency is not further enhanced. We appreciate the reviewer for encouraging us to include these experimental results, as they further clarify that the gains of resolution extrapolation are primarily in image quality.
>
> | Different Resolution | FID | FID_crop | Clip Score |
> | --- | --- | --- | --- |
> | 1024×1024 | 63.47 | 63.95 | 28.52 |
> | 2048×2048 | 57.69 | 55.59 | 28.49 |
> | 3072x3072 | 53.21 | 52.91 | 28.46 |
> | 4096x4096 | 53.75 | 53.01 | 28.41 |
>
> In addition, the Black Forest Lab Team claims that Flux.1-dev can directly generate images at 2048×2048, but it has been noticed that the generation quality at 2048×2048 is not particularly good enough. Therefore, we also compared the results of using Flux.1-dev with its default settings against Flux.1-dev enhanced with I-Max at the 2048×2048 resolution. I-Max also achieves a significant margin of improvement, indicating its practical value in real-world applications.
>
> | Different Step Numbers | FID | FID_crop | Clip Score |
> | --- | --- | --- | --- |
> | Default Setting | 59.10 | 75.53 | 28.29 |
> | Enhanced with I-Max | 53.75 | 53.01 | 28.41 |
>
> As for the comparison between different low-resolution guidance approaches, we report the results in the table below. All metrics show strong consistency with the GPT-4o preference evaluation in Fig. 7, demonstrating the superiority of the proposed projected flow approach.
>
> | Different Guidance | FID | FID_crop | Clip Score |
> | --- | --- | --- | --- |
> | Projected Flow | 53.75 | 53.01 | 28.41 |
> | SDEdit | 54.89 | 55.76 | 28.32 |
> | Skip Residual | 53.94 | 53.89 | 28.40 |
> | Time-aware Scaled RoPE | 54.93 | 54.37 | 27.96 |
>
> Once again, we thank the reviewers for emphasizing the importance of using multiple quantitative metrics for evaluation. We will revise the paper and include these experimental results. Regarding the reviewers' suggestion to evaluate on low-resolution diffusion models and datasets, we acknowledge that this is an aspect often overlooked by current resolution extrapolation work. Due to time constraints, during the rebuttal phase, we mainly focus on supplementing experiments based on the existing base model. However, we will consider including results on other models in the camera-ready version.
>
> **W2. Exploring the impact of different low-pass filters.**
>
> Good point! In the paper, we by default use low-pass filtering based on Discrete Wavelet Transform. In the table below, we report the evaluation results for three types of filters: (1) Discrete Wavelet Transform, (2) Fast Fourier Transform, and (3) low-pass filtering based on downsampling-upsampling. We observe that different filters do not have a significant impact on the performance of I-Max, with DWT showing slightly better overall performance. We will include these results and the corresponding discussion in the camera-ready version.
>
> | Different Projection | FID | FID_crop | Clip Score |
> | --- | --- | --- | --- |
> | DWT | 53.75 | 53.01 | 28.41 |
> | FFT | 53.64 | 53.56 | 28.37 |
> | Downsample-Upsample | 53.98 | 53.57 | 28.43 |
>
> *[1] Du, Ruoyi, et al. "Demofusion: Democratizing high-resolution image generation with no $$$", CVPR, 2024.*
>
> *[2] Yang, Zhuoyi, et al. "Inf-dit: Upsampling any-resolution image with memory-efficient diffusion transformer", ECCV, 2024.*

---

> ### Author Response · Authors · 2024-12-02
> **Looking forward to your response**
>
> Dear Reviewer jzk2,
>
> Considering the discussion period is about to end, we sincerely look forward to your replies. We hope that we have addressed your concerns; if not, we are happy to listen to the follow-up questions. Thanks!
>
> Bests,
>
> Authors of Submission 2180

---

### Meta-Review · Area_Chair_M42o · 2024-12-24

**Metareview:**

This paper proposes a tuning-free method to enhance the spatial resolution of text-to-image generation models, grounded on rectified flow transformers. The proposed I-Max framework is thoughtfully designed to improve stability during resolution extrapolation, addressing core questions such as "how to guide" and "how to infer." The paper demonstrates the authors' deep understanding of the tuning-free resolution extrapolation task and their effort to achieve stable performance for a more scalable T2I framework based on RFT, diverging from the conventional U-Net structure.

However, the authors appear to have overlooked the importance of providing comprehensive evaluations and direct comparisons to effectively persuade reviewers. Reviewers primarily highlighted two issues: the lack of quantitative evaluations and insufficient comparisons with state-of-the-art (SOTA) methods. In response to the former, the authors presented FID and FID-crop scores to validate their approach using GPT-4o. For the latter, they provided limited results during the rebuttal stage.

Given the nature of this work, visual comparisons are crucial, but as the authors acknowledged, presenting such results in the rebuttal stage is almost impossible. Consequently, reviewers struggled to make proper judgments based solely on text-based feedback. Tuning-free add-ons are known to risk introducing artifacts and irrelevant details. Successful works in this area, such as Demofusion, typically include detailed visual comparisons and even interactive tools for demostration. Additionally, the reviewers' request for failure cases was not fully addressed, likely due to the same limitation of not being able to include images.

After double-checking the appendix, AC confirmed that the authors did not provide sufficient visual results, direct comparisons with other SOTA methods, or failure case analyses. Given that comprehensive evaluations and direct comparisons are crucial for assessing the effectiveness of such a tuning-free tool, the AC believes this paper falls slightly below the acceptance threshold and recommends rejection in its current form.

**Additional Comments On Reviewer Discussion:**

The AC acknowledges private communications from the authors. One possible reason for the authors’ hesitance to engage in further discussion might be the difficulty of providing fair evaluations without visual results. While Reviewer Jtf5 is slightly positive about the paper, rating it marginally above the acceptance threshold, the other three reviewers are mildly negative toward its acceptance. The AC, having direct experience with tuning-free resolution extrapolation for T2I models, agrees that comprehensive evaluation—both qualitative and quantitative—is crucial for such works. Authors are strongly encouraged to resubmit this work to a future venue after significantly enriching the experimental section to provide a more thorough evaluation (all in the review package that reviewers can access directly).

---

### Decision · Program_Chairs · 2025-01-22

Reject